# Impact of Exercise Manual Program on Biochemical Markers in Sedentary Prediabetic Patients: A Randomized Controlled Trial

**DOI:** 10.3390/medicina61020190

**Published:** 2025-01-22

**Authors:** Sana Hafeez, Syed Shakil Ur Rehman, Saima Riaz, Imran Hafeez, Zarwa Hafeez, Hassan Mumtaz

**Affiliations:** 1Faculty of Rehabilitation and Allied Health Sciences, Riphah International University, Lahore 54000, Pakistan; sanahpt2@gmail.com (S.H.); shakil.urrehman@riphah.edu.pk (S.S.U.R.); 2Department of Physical Therapy and Rehabilitation Sciences, University of Management and Technology, Lahore 54000, Pakistan; 3Ayesha Bakht Institute of Medical Sciences, Lahore 54000, Pakistan; saima.riaz@abims.edu.pk; 4Children Hospital, University of Child and Health Sciences, Lahore 54000, Pakistan; radicular12@gmail.com; 5National Hospital and Medical Center, Lahore 54000, Pakistan; zarwahafeez123@gmail.com; 6Data Analytics, BPP University, London W12 8AA, UK

**Keywords:** prediabetes, exercise manual, HbA1c, insulin sensitivity, lipid profile, exercise interventions

## Abstract

*Background and Objectives*: Prediabetes is a medical disease characterized by elevated blood sugar levels that exceed normal levels but do not meet the criteria for a diagnosis of diabetes mellitus. This study aimed to assess the impact of structured exercise manual interventions on the biochemical markers of sedentary prediabetic patients over sixteen weeks. *Materials and Methods*: A sixteen-weeks randomized controlled trial was conducted to assess the impact of an exercise-based manual program on biochemical markers, such as HbA1c, insulin sensitivity measures, and lipid profiles, in sedentary individuals with prediabetes. The Riphah Rehabilitation Center in Lahore, Pakistan, was the site of the trial. In this investigation, 126 individuals with prediabetes were randomly assigned to three groups: control, unsupervised, and supervised. The RCT was completed by 36 participants in each group after a 16-weeks intervention in the supervised and unsupervised groups, as well as a follow-up in the control group. An activity-based exercise manual that included dietary guidelines, educational materials, and an exercise routine was followed by both the supervised and unsupervised groups. The exercise interventions included both aerobic and resistance components. *Results*: The results indicated that the supervised group exhibited a substantial increase in insulin sensitivity, lipid profiles, and glycemic control when contrasted with the unsupervised and control groups. Significant improvements were observed in key biochemical parameters, including fasting blood levels (supervised as compared to unsupervised and control, respectively, the mean difference was 12.82 mg/dL vs. 11.36 mg/dL vs. 0.09 mg/dL *p* > 0.001), HbA1c (supervised as compared to unsupervised and control groups, respectively, the mean difference was 0.67% vs. 0.69% vs. 0.13% *p* < 0.001), and lipid profile (triglycerides (mean difference 0.25 mmol/L, 0.08 mmol/L, 0.11 mmol/L *p* < 0.001); LDL (mean difference 19.31 mg/dL, 10.51 mg/dL, 2.49 mg/dL *p* < 0.001); HDL (mean difference −12.68 mg/dL, −8.03 mg/dL, −1.48 mg/dL *p* < 0.001)). In comparison to the unsupervised and control groups, the insulin sensitivity parameters also demonstrated a modest improvement in the supervised group. The supervised group exhibited the greatest benefits from exercise among the groups that received exercise interventions. *Conclusions*: The present investigation demonstrated the significance of including structured physical activity into the regular routine of individuals with prediabetes, to decelerate the advancement of prediabetes to type 2 diabetes mellitus (T2DM). The current study emphasizes the essential role of structured exercise routines in the control of prediabetes and suggests that monitoring enhances the adherence and effectiveness of lifestyle interventions.

## 1. Introduction

The American Diabetes Association introduced the term “prediabetes” in its treatment guidelines over two decades ago, superseding the terms “impaired glucose tolerance” (IGT) and “impaired fasting glucose” (IFG) [1]. Intermediate hyperglycemia is defined as the presence of at least one of the following: IGT (2-h glucose 7.8–11.0 mol/L [140–199 mg/dL]), IFG (fasting glucose 6.1–6.9 mmol/L [110–125 mg/dL]), or modestly elevated HbA1c (5.6–6.5) levels. It is indicative of prediabetes [2,3]. IGT is typically defined by insulin resistance in muscle and reduced glucose uptake, whereas IFG is typically characterized by insulin resistance in the liver and excessive hepatic glucose production [4,5,6]. Although prediabetics may not exhibit symptoms that are typically associated with hyperglycemia, they are at a high risk of developing diabetes, with up to 50% of them progressing to diabetes within five years [7].

The progression of the epidemic of type 2 diabetes mellitus (T2DM) persists. It is anticipated that the global burden of T2DM will increase to 578 million in 2030 and then to 700 million in 2045 [8]. The prevalence of prediabetes in Pakistan is significantly higher than the previous estimate [9]. In the year 2021, according to the 10th iteration of the International Diabetes Federation (IDF) Atlas, Pakistan was the third most prevalent country in the world for diabetes, following China and India [10]. In 2022, the prevalence of diabetes in Pakistan was dominating, with an estimated 26.7% of adults affected. This resulted in a total of approximately 33,000,000 cases [11]. Biochemical variables show improvements in reducing glucose levels after exercise. HbA1c is a critical indicator for evaluating sustained changes in glucose metabolism, due to the interventions reflecting long-term glycemic control over the preceding 2–3 months. FBG provides immediate blood glucose regulation, helping to improve short-term metabolic responses. In diagnostic criteria for both prediabetes and diabetes, these markers are essential for tracking the progression or reversal of prediabetes. Triglycerides and LDL (low-density lipoprotein), often elevated in prediabetic and diabetic conditions due to a dysregulated lipid metabolism, are key markers of cardiovascular risk. HDL (high-density lipoprotein), the “good cholesterol”, is inversely associated with cardiovascular risk. HOMA-IR (homeostatic model assessment of insulin resistance), a central feature of prediabetes, quantifies insulin resistance. HOMA-B assesses pancreatic beta-cell function, crucial for understanding how exercise interventions preserve or improve insulin secretion capacity.

Prediabetes is associated with a 1.5-fold increased risk of heart and blood vessel disease, including high blood pressure, stroke, and heart attack, in addition to an elevated risk of diabetes [12]. Prediabetes can be prevented from progressing to type 2 diabetes through interventions such as weight loss, physical activity, and healthy dietary choices [13,14]. Within five years, it is probable that 15% to 30% of individuals with prediabetes will develop type 2 DM without intervention [15]. Prediabetes is considered the most critical stage of the preventative process because it provides a window of opportunity to treat T2DM, as the patient’s condition is reversible at this stage [16]. Lifestyle modifications are a critical priority for various health organizations and serve as a preventative measure against the progression of diabetes [17,18,19,20]. The lifestyle intervention program is determined to be effective; however, it underscores the obstacles associated with sustained adherence [21]. Physical activity emerges as the pivotal axis amidst multifarious lifestyle modifications [22]. It is essential to acknowledge that physical activity is the foundation, as its profound effects on metabolic pathways and neuroendocrine systems become increasingly apparent.

Physical activity (PA) encompasses “all movement during leisure time” and is defined as “any bodily movement produced by skeletal muscles that requires energy expenditure” [23]. Structured physical activity involves all body movement that increases energy use while performing exercise. To prevent or delay type 2 diabetes mellitus, exercise plays an important role as it lowers blood glucose levels, lessens the cardiovascular risk factors, plays an important part in weight loss, and improves well-being [24]. Diabetes types, activity types, and the presence of diabetes-related complications are varied, with challenges related to blood glucose management. Physical activity and exercise recommendations, therefore, should be tailored to meet the specific needs of each individual. The level of PA was calculated: high, for those individuals who had vigorous activity at least three days a week; moderate, for those who had three or more days of intense activity of at least 20 min per day, or five or more days of moderate activity, or walking for at least 30 min per day; and a low level for those who did not meet the above criteria. Both a high and moderate level of physical activity is recommended for improving glucose levels. Regular physical activity is recommended for patients with different metabolic diseases including diabetes, dyslipidemia, and metabolic syndrome. Glucose intolerance, hypertension, dyslipidemia, and central obesity is an overwhelming health problem across the globe. Regardless of the molecular pathways, regular exercise, especially a program with a combination of aerobic and resistance training, can improve most conditions related to metabolic syndrome [25]. The fourth risk factor for global mortality is physical inactivity, and it is estimated to be the main cause of 27% of diabetes cases. Physical activity can be mild, moderate, or intense; the higher the performance, the greater the results shown. On the other hand, it is convenient to point out that regardless of the PA performed, spending a lot of time in sedentary behaviors can additionally increase health risks [26].

The most commonly used modalities of exercise therapy are aerobic exercise and resistance training. Both exercise regimes are equally good and have their own benefits for muscles and glucose monitoring. Studies have shown that aerobic exercises can stabilize blood pressure, enhance lipoprotein density, and enhance cardiovascular function, thus decreasing the risk of cardiovascular disease. On the other hand, resistance training is primarily beneficial as it improves the function and mass of skeletal muscles, as well as energy expenditure and metabolic activity in muscle tissue. In some studies, combined training has been shown to improve lipid and vascular function more effectively than AE or RT alone. The evidence suggests that aerobic exercise and resistance training have beneficial effects on various participants [27].

Both aerobic and resistance exercise within a single session have a positive impact on body mass and composition, as well as enhanced glucose and lipid metabolism. The benefits of combined training in improving glycemic management and lipid parameter health in obese adults are proven by many research studies. Current ADA guidelines advocate a combination of aerobic and resistance training, which may be the most beneficial exercise modality for regulating lipid profiles and glucose levels [28].

Regular exercise training reduces A1C, triglycerides, blood pressure, and insulin resistance in individuals with prediabetes. Resistance training is an exercise intervention that can help to reduce the insulin resistance and improve blood glucose level [29].

A systematic review and meta analysis suggested that resistance training shows major changes on strength, body composition and psychosocial status of overweight children and adolescents.

Alternatively, resistance training promotes the rapid enhancement of skeletal muscle oxidative capacity, insulin sensitivity, and glycemic control in adults with prediabetes. For low muscular strength and an accelerated decline in muscle strength and functional status, diabetes is an independent risk factor [30]. The health benefits of resistance training for all adults include improvements in muscle mass, body composition, strength, physical function, mental health, bone mineral density, insulin sensitivity, blood pressure, lipid profiles, and cardiovascular health. Performing resistance exercise first results in less hypoglycemia than when aerobic exercise is performed, when resistance and aerobic exercise are undertaken in one exercise session. For individuals with type 2 diabetes, the benefits of a combination of aerobic and resistance training include improvements in glycemic control, insulin resistance, fat mass, blood pressure, strength, and lean body mass [31].

Prediabetic individuals should engage in at least 150 min of physical activity per week, as advised by the American Diabetes Association (ADA) [31,32,33]. Oral glucose tolerance, fasting blood glucose, and glycated hemoglobin (HbA1C) were all improved in individuals with prediabetes who participated in a training program [34,35]. A meta-analysis of 16 randomized controlled trials found that individuals with prediabetes who received lifestyle interventions, including increased PA levels, had a lower rate of progression to T2D after one (RR = 0.46, 95% CI = 0.32–0.66) and three years of follow-up (RR = 0.64, 95% CI = 0.53, 0.77) [18]. In recent years, the absence of physical activity has become a significant public health concern. According to statistics, 60.1% of the Pakistani population is not engaging in physical activity to acquire its health benefits [36]. There are some cultural barriers that can be a hinderance to engaging in physical activity

Gender roles and societal norms in Pakistan may restrict outdoor physical activity, particularly for women, due to concerns about safety and privacy, the limited availability of female-only fitness facilities, and a lack of culturally acceptable exercise attire. Family responsibilities are often prioritized over individual health, reducing time availability for structured exercise. This exercise manual provides a structured exercise program that can be easily performed indoors with no equipment needed, and also it is cost effective. Community-based programs and group exercises have shown promise in overcoming barriers by creating a sense of collective motivation. This exercise manual program is about community awareness and the prevention of diabetes. Increasing awareness through culturally tailored health education campaigns can also serve as a facilitator.

Consequently, an exercise-based manual program was developed to assess the combined impact of diet, education, and exercise interventions on biochemical parameters, including fasting blood glucose, HbA1c, lipid profile, HOMA-B, and HOMA-IR, in individuals with prediabetes. In the literature, there are studies that compare supervised with unsupervised groups and supervised with control groups. But there is a gap in the literature about the comparison of supervised, unsupervised, and control groups [32].

The framework of the exercise manual is based on content that can be easily performed with no hassles and cost effectiveness, as cultural norms in Pakistan show a hindrance for outdoor exercise. The purpose of adding an unsupervised group is to motivate the participants in performing exercises on their own. It shows some novelty in our work that we added the comparison of supervised, unsupervised, and control groups. The objective of this manual is to address a critical lackness in the healthcare system by developing and executing exercise interventions that are specifically designed for individuals with prediabetes. This exercise-based manual contributes to the prevention of the escalating prevalence of diabetes and the enhancement of public health outcomes by addressing modifiable risk factors and the potential of lifestyle modifications, particularly through the implementation of an exercise protocol. This manual, which is exercise-based, aids in the dissemination of information regarding prediabetes at the community level and offers an exercise protocol to mitigate hyperglycemia in the intended demographic.

## 2. Material and Methods

### 2.1. Study Design and Setting

This prospective 16-weeks double-blinded randomized controlled trial was conducted from April 2022 to November 2023 at the Riphah Rehabilitation center, Lahore. The clinical trial protocol was approved by the Research and Ethics Committee, Riphah College of Rehabilitation and Allied Health Sciences, Riphah International University, Lahore, Pakistan (reference# RFC/RCR & AHS 21/1103) and was conducted in accordance with the ethical standards set by the Helsinki Declaration of 1975. The clinical trial (NCT04949958) was registered in the Clinical Trials.gov Protocols Registration and Results System on 28 April 2021. The study followed the CONSORT 2010 Statement (Consolidated Standards of Reporting Trials) [37].

### 2.2. Sample Size Calculation

Pilot study used to determine sample size calculation. Effect size 0.27 yielded from pilot study. Sample size of 105 calculated with effect size 0.27, power 80%, and margin of error 5%. After inclusion of 20% attrition rate, sample size was determined as 126, with 42 participants allocated to each group.

### 2.3. Study Participants

The consenting participants in this randomized controlled trial were Pakistani sedentary adults of both genders, aged 18 to 40. Willing participants were enrolled if they were overweight or obese by body mass index (BMI of ≥23.0 kg/m^2^) [38,39] or had a fasting plasma glucose of 100–125 mg/dL or an HbA1c level of 5.7–6.4% [40,41]. They were recruited through print and visual media campaigns, as well as medical camps. Prior to engaging in any activity, all participants completed the PAR-Q + Physical Activity Readiness Questionnaire.

Excluded participants were pregnant women, individuals with established type 1 or T2DM, those on any herbal medications for weight loss therapy, individuals who were involved in other weight management or structured exercise programs in the past six months, and individuals with chronic medical conditions, such as advanced arthritis, renal, hepatic, and cardiac complications.

An assessor conducted an initial assessment of all the individuals to ensure that they met the inclusion criteria. The participants were subsequently assessed twice during the course of the study: at the commencement and at the conclusion of the sixteen-weeks intervention. All participants were instructed on the exercise-based manual and were introduced to the study prior to their participation. Each participant was required to provide informed consent prior to participating in the study.

### 2.4. Randomization and Blinding

A simple randomization process using a computer-generated randomization table was used. Participants who signed the informed consent form and agreed to participate in the study were randomly assigned to one of three treatment groups. All personnel involved in the study, including the assessor and participants, were blinded to the treatment, with the exception of the therapist who administered it. The participants were randomly allocated to the supervised, unsupervised, or control group (Figure 1).

### 2.5. Study Groups and Interventions

A total of 126 individuals with prediabetes were enrolled and randomly assigned to one of three categories. Each group consisted of 42 individuals with prediabetes.

### 2.6. Supervised Group

This group (A) consisted of participants who adhered to aerobic and resistance exercises under the supervision of trained staff, along with a homogeneous diet plan, for a period of four months. They started their exercises with a 10 min warm up, followed by 40 min aerobic and resistance exercises regime and 10 min cooldown period with moderate intensity, using a 70–75% HR peak and 40 s recovery time.

### 2.7. Unsupervised Group

This group (B) received the aerobic and resistance exercises after the trial’s induction and were instructed in their use. A training session was given to the unsupervised group before starting the interventions, so they would know about the correct pattern of exercises. Participants recorded their exercise routine in a logbook during home training days. Participants were asked to record (a) the activity performed, (b) the minutes spent engaging in the activity, (c) the frequency of the session, and (d) how hard the session was. They were permitted to follow the manual at home and monitor their performance using the diary method for the same period. Measurements were taken at the start and end of the 16 weeks.

### 2.8. Control Group

The control group, which consisted of prediabetic individuals, was designated as group C. They received dietary advice and were provided with a notebook in which to mention their routine. They followed the same diet as the other groups. Measurements were taken at the start and end of the 16 Weeks [42].

### 2.9. Exercise Manual Program

The exercise-based manual for prediabetic individuals was divided into five primary sections: prediabetes education, health precautions, diet plans, exercise education, and exercise delivery [22,43]. The exercise delivery component consisted of aerobic and resistance elements, with 10 min of jogging prior to commencing the exercise and 10 min of strolling to cool down after the exercise [44].

### 2.10. Aerobic Component

These exercises performed in a sequence: walk on spot, standing arm circles, walk on spot, wall pushups, walk on spot, standing crisscross crunches, walk on spot, jumping jacks, walk on spot, side lunges, walk on spot, and high knees. They started the exercises with a 10 min warm up, followed by aerobic exercise regime and 10 min cooldown period with moderate intensity, using a 75% HR peak and 40 s recovery time. Participants performed 2 sets with 10 repetitions and a 40 s rest interval in between each set. The exercise was performed 3 times/week [45].

### 2.11. Resistance Component

The resistance training emphasized the elbow plank, which was performed twice a week for 15 s with 4 repetitions. The exercise was performed along with the 1st and 3rd day of the aerobic component.

#### 2.11.1. Exercise Progression

Exercise progression is necessary to challenge your body capacity. Any change in exercise prescription means to change into duration that make exercise longer, change into intensity that make exercise more intense. Here is listed of exercise progression of both aerobic and resisted.
Aerobic (3 times/week)Resistance (2 times/week)1–4 weeks: 2 sets with 10 repetitions, 40 s1–2 weeks:15 s hold with 4 reps5–10 weeks: 3 sets with 15 repetitions, 40 s rest3–4 weeks: 30 s hold with 8 reps11–16 weeks: 4 sets with 20 repetitions, 30 s rest5–8 week: 45 s hold with 12 reps9–12 weeks: 1 min hold with 16 reps13–16 weeks: 1 min 30 s hold with 20 reps

The diet plan for all three groups (supervised, unsupervised, and control) was the same. The diet plan focused on balanced nutrition, including low glycemic index foods, portion control, and proper hydration.

#### 2.11.2. Outcome Variables

The primary objective of this study is to evaluate the impact of a structured exercise manual on the biochemical health of sedentary prediabetic individuals. The selected markers directly measure the outcomes linked to this objective: Glycemic markers (HbA1c, FBG) assess the core metabolic improvement in glucose control. Lipid profiles (triglycerides, LDL, HDL) monitor the cardiovascular benefits of the intervention. HOMA-IR and HOMA-B address the mechanistic aspects of insulin resistance and pancreatic function, highlighting physiological adaptations to exercise. The results demonstrate that supervised interventions led to significant improvements in these markers, substantiating their relevance and the interventions’ effectiveness in mitigating progression to type 2 diabetes.

Biochemical characteristics were measured by trained staff at the center at the time of recruitment (baseline), and after 16 weeks of intervention. An overnight fasting blood sample was collected in a red-top vacutainer from each participant and centrifuged using a standard tabletop centrifuge for the separation of serum. The serum samples collected were aliquoted in multiple 1.5 μL Eppendorf tubes and transported immediately for biochemical analysis of their fasting blood glucose and lipid profile, using an automated biochemical analyzer (Konelab 18, Thermo-Fischer, Vantaa, Finland). A standard glucose–oxidase–peroxidase (GOD-POD) method was used to assess fasting glucose (Hitachi 7020, Tokyo, Japan). HbA1C levels were measured using a Bio-Rad point-of-care device, which performs the analysis using boronate affinity chromatography (Bio-Rad Laboratories, Hercules, CA, USA). Insulin resistance was calculated by means of the homeostasis model assessment of insulin resistance (HOMA-IR) method [46]. The related formulas for calculation are as follows:HOMA-IR = [fasting plasma insulin (µU/mL) × fasting plasma glucose (mg/dL)]/405.HOMA-B = 20 × fasting insulin (µU/mL)/[fasting plasma glucose (mg/dL) − 63]

### 2.12. Statistical Analysis

The biochemical data collected for the two timepoints for each participant of all the groups were compiled and analyzed using SPSS version 25. The data were checked for normality using the Shapiro–Wilk test. The data at baseline and 16 weeks were presented as the mean ± standard deviation for normally distributed variables and *n* (%) for categorical variables.

Intervention effects giving the adjusted mean difference between groups were calculated using a Mixed-Effect Model, and those within the groups were calculated with a paired *t*-test, with fasting blood glucose, HbA1c, triglycerides, LDL–cholesterol, HDL–cholesterol, HOMA-IR, and HOMA-B. A *p* < 0.05 was considered significant. The clinical reference values of all biochemical variables are listed (Table 1).

## 3. Results

### 3.1. Participant Demographics and Biochemical Characteristics

For the current RCT, a total of 514 individuals were enrolled; only 126 participants were randomly divided into three groups. Each group had 42 participants based on the 90% desired power of the study. A random number table was used to divide the participants into three groups, initially with 42 people in each group. After 16 weeks of intervention, 108 participants completed the study, with 36 participants in each group.

Table 2 provided a comprehensive overview of the demographic characteristics of the participants across the three distinct groups in the study: the supervised group, the unsupervised group, and the control group. The data encompass gender distribution, as well as mean values for age, weight, height, and body mass index (BMI), with accompanying standard deviations (SD) to indicate variability within each group.

The gender distribution across the groups showed a higher proportion of males in the unsupervised group (55.6%) compared to the supervised group (41.7%) and the control group (50%). Females comprised 58.3% of the supervised group and 50% of the control group, slightly more than in the unsupervised group (44.4%). However, a statistical analysis indicated no significant differences in gender distribution among the groups (males: chi square: χ^2^ = 1.7, *p* = 0.42; females: χ^2^ = 1.3, *p* = 0.58). The average age of participants was similar across the groups, with the supervised group averaging 32.25 ± 5.69 years, the unsupervised group at 30.58 ± 3.30 years, and the control group at 30.75 ± 4.14 years. The analysis suggested that age differences were not statistically significant (*p* > 0.05).

Mean weight and BMI showed no significant differences between the groups, with the supervised group having a mean weight of 85.03 ± 4.7 kg and a BMI of 30.58 ± 1.25 kg/m^2^, while the unsupervised and control groups had similar weight and BMI values. A weight comparison yielded a *p* > 0.05, indicating no significant difference, and a BMI comparison also showed a *p* > 0.05. Height was the only variable showing a statistically significant difference between the groups, with the control group being the tallest on average (170.08 ± 4.43 cm) compared to the supervised group (166.16 ± 3.60 cm) and the unsupervised group (168.58 ± 4.42 cm) (*p* < 0.05).

The demographic data, except for height, showed no significant differences between the baseline data of the experimental and control groups, suggesting that any outcomes of the study are unlikely to be attributable to demographic variance. Chi-square tests and one-way analyses of variance (ANOVAs) indicated that the baseline data on each dependent variable among the three groups were homogeneous.

The baseline data of the biochemical variables collected from participants before the initiation of an exercise manual intervention are provided in Table 1. The table presents the mean values and the standard deviation (SD) for each biochemical variable, including fasting blood glucose (FBG), hemoglobin A1c (HbA1C), triglycerides (TGs), low-density lipoprotein (LDL), high-density lipoprotein (HDL), homeostatic model assessment for beta-cell function (HOMA-B), and homeostatic model assessment for insulin resistance (HOMA-IR). The F-statistics and corresponding *p*-values are provided to denote the significance of the differences among the groups.

The glycemic indicators showed a significant difference for the baseline data. The FBG levels were slightly higher in the unsupervised group compared to the supervised group, while lower than the control group, with a significant difference of (F = 7.9, *p* < 0.05). The HbA1C levels were also significantly different between the three groups (F = 7.32, *p* = 0.001).

In the lipid profile, the baseline values of triglycerides and LDL among the groups were significant (F = 786.3, *p* < 0.05 and F = 5.8, *p* < 0.05, respectively). However, the HDL levels were not significantly different (F = 18.55, *p* > 0.05).

HOMA-B values were (69.03 ± 1.20) in the supervised group, 71.46 ± 2.31 in the unsupervised group, and 79.28 ± 3.55 in the control group, indicating a statistically significant difference (F = 18.46, *p* < 0.05). Finally, HOMA-IR levels were similar across all groups, with no significant differences observed (supervised group: 1.00 ± 0.04; unsupervised group: 0.99 ± 0.02; control group: 1.03 ± 0.02; F = 494.02, *p* > 0.05).

The biochemical profile suggested that while some differences in metabolic indicators are evident between the groups, particularly in glucose and lipid metabolism, insulin resistance measured by HOMA-IR was comparable across the groups. The significance of these findings lies in understanding the potential impacts of supervision in lifestyle interventions and their effects on metabolic health.

### 3.2. Comparison of Biochemical Indicators at Baseline and 16 Weeks of Intervention in Supervised Group

Table 3 showed a comparison of various biochemical indicators at baseline and after a 16-week intervention for the supervised group. The supervised group followed the exercise manual under the supervision of trained staff. The parameters measured include fasting blood glucose (FBG), hemoglobin A1c (HbA1C), triglycerides (TGs), low-density lipoprotein (LDL), high-density lipoprotein (HDL), and homeostatic model assessment for beta-cell function (HOMA-B).

From baseline to 16 weeks, there were statistically significant reductions in FBG, HbA1C, TGs, LDL, and HOMA-B values, as indicated by the *p*-value of <0.001 for each, which denotes a probability of less than 0.1% that these results are due to chance, thereby surpassing the threshold for statistical significance (*p* < 0.001). The Cohen’s d values suggest large effect sizes for FBG (5.347), TGs (3.945), LDL (2.653), and HOMA-B (6.697), with the largest effect observed in HOMA-B and the smallest yet considerable effect in HbA1C (0.705). Conversely, HDL levels showed a significant increase over the intervention period, with the Cohen’s d indicating an extremely large effect size (21.56). The mean HOMA-IR increased from baseline to 16 weeks, and the paired *t*-test indicates a highly significant difference in HOMA-IR values over this period, with a very large effect size (Cohen’s d). This suggests that the intervention had a strong impact on HOMA-IR levels.

Overall, these results demonstrate that the exercise intervention was highly effective in improving these biochemical parameters among the participants, with the most pronounced effects on insulin resistance as measured by HOMA-B and on the improvement in HDL levels, commonly associated with reduced cardiovascular risk. The significant decrease in FBG, HbA1C, triglycerides, and LDL and increase in HDL and HOMA- B suggest that exercise intervention under supervision can have a profound impact on metabolic health.

### 3.3. Comparison of Biochemical Indicators at Baseline and 16 Weeks of Intervention in Unsupervised Group

The outcomes of interventions on various biochemical indicators in a group of individuals undertaking exercise-based manual interventions at home are presented in Table 4. The parameters measured include fasting blood glucose (FBG), glycated hemoglobin (HbA1c), triglyceride, low-density lipoprotein (LDL), high-density lipoprotein (HDL), homeostatic model assessment for insulin resistance (HOMA-IR), and beta-cell function (HOMA-B).

The means (M) and standard deviations (SDs) at baseline and after 16 weeks are provided for each variable, alongside the t-statistic, *p*-value, and Cohen’s d for the change over time. Notably, all variables exhibit a significant change, with *p* < 0.001.

FBG levels decreased from a mean of 107.68 mg/dL to 96.32 mg/dL, indicating an improvement in blood glucose regulation. This change is underscored by a large effect size (Cohen’s d = 5.60). Similarly, HbA1c, a marker of long-term glycemic control, showed a reduction from 6.11% to 5.42%, which suggests better glycemic control over the preceding months. The effect size for this change is moderate (Cohen’s d = 0.41).

Triglyceride experienced a modest reduction from 1.95 mmol/L to 1.86 mmol/L, with a small effect size (Cohen’s d = 0.18). However, LDL decreased significantly from 125.11 mg/dL to 114.60 mg/dL, with a large effect size (Cohen’s d = 1.72), indicating an improved lipid profile. HDL increased from 36.33 mg/dL to 44.35 mg/dL, which is favorable for cardiovascular health. This change also had a large effect size (Cohen’s d = 2.33).

HOMA-B showed an increase from 71.46 to 77.46, suggesting an improvement in insulin secretion, with a large effect size (Cohen’s d = 2.87). Finally, HOMA-IR showed an increase from 0.992 to 1.051, representing a change in insulin resistance. The effect size for this was moderate (Cohen’s d = 1.94). The interventions led to significant improvements across various metabolic parameters over the 16-week period. The findings suggest the potential efficacy of at-home exercise interventions for improving metabolic health, though the increase in HOMA-IR may warrant a more nuanced interpretation.

### 3.4. Comparison of Biochemical Indicators at Baseline and 16 Weeks in Control Group

The Table 5 summarizes the impact of non-intervention on several biochemical markers in a control group over a 16-week period. The metrics considered include fasting blood glucose (FBG), glycated hemoglobin (HbA1c), triglycerides, low- and high-density lipoproteins (LDL and HDL, respectively), and indicators of insulin function and resistance (HOMA-B and HOMA-IR).

Statistically, no significant change was detected in FBG and HbA1c levels, as indicated by *p*-values well above the 0.001 threshold. However, substantial increases in triglyceride levels were observed, with a very large effect size (Cohen’s d = 2.23), suggesting that despite the absence of a targeted intervention, lipid metabolism may have been adversely affected in this cohort.

Interestingly, LDL levels demonstrated a slight decrease, albeit with a small effect size (Cohen’s d = 0.44), which might not be clinically significant. HDL levels saw a modest increase with a medium effect size (Cohen’s d = 0.50), which could be interpreted as a positive change in cardiovascular risk factors.

Notably, HOMA-B levels, reflecting the beta-cell function of the pancreas, showed a slight increase with a large effect size (Cohen’s d = 2.67). This could suggest that pancreatic function was compensating for increasing insulin resistance or other metabolic pressures. The most striking finding is the marked increase in HOMA-IR, with a mean going from 1.034 to 1.262. This change was highly significant (*p* < 0.001), with an extremely large effect size (Cohen’s d = 9.62), indicating that insulin resistance significantly worsened in this group over the 16 weeks.

The data from the control group suggested that in the absence of an intervention, individuals experienced notable negative shifts in lipid metabolism and insulin resistance. The changes in triglyceride levels and HOMA-IR suggest worsening metabolic health, underscoring the potential benefits of exercise interventions in maintaining or improving metabolic function.

### 3.5. Intervention Effects on Biochemical Indicators

Table 6 presents the effects of a 16-week intervention on various biochemical indicators across three groups: supervised, unsupervised, and control. The key findings are summarized as follows:

Both supervised and unsupervised groups showed significant reductions in HbA1C and FBG levels post-intervention, indicated by low *p*-values (<0.001 < 0.001 < 0.001) and large negative Cohen’s d values, which suggest the substantial effect of the intervention. The supervised group showed a greater reduction than the unsupervised group, indicating a potentially stronger effect when the intervention is supervised. The control group showed only minor changes in HbA1C and FBG, with smaller effect sizes and *p*-values still below the significance threshold for HbA1C, suggesting a lesser natural variation over time.

Significant improvements were observed in insulin resistance indicators (HOMA-B and HOMA-IR) for both the supervised and unsupervised groups, with large effect sizes and high statistical significance. The control group exhibited minimal or non-significant changes, supporting the intervention’s effectiveness in enhancing insulin sensitivity.

The intervention positively impacted lipid levels, especially in the supervised group, where triglycerides and LDL decreased, while HDL showed a beneficial increase. The unsupervised group also demonstrated significant, albeit slightly smaller, changes. In contrast, the control group showed minimal or non-significant changes in lipid levels.

### 3.6. Mixed-Effects Model Results

Table 7 presents the results of the Mixed-Effect Model analysis for each biochemical variable, with estimates for the baseline mean, the effect of time (pre- and post-intervention), group comparisons, and time-by-group interactions.

The intervention had a significant effect on key biochemical indicators. The supervised group showed a greater improvement in FBG during the intervention period, reinforcing the superiority of supervised exercise for improving fasting blood glucose levels. Similarly, the supervised group had significantly lower HbA1c levels compared to the control group at baseline, and previous analyses have shown that the supervised intervention had a stronger impact on reducing HbA1c over time, highlighting the effectiveness of supervised exercise for long-term blood sugar regulation.

Significant differences in all pairwise comparisons were observed in triglycerides, while in case of LDL only the supervised vs. unsupervised comparison was significant. However, there were significant differences for both control vs. supervised and control vs. unsupervised but no significant difference between the exercise groups observed in HDL. HOMA-B: All comparisons were significant. HOMA-IR: Significant differences between control vs. supervised and control vs. unsupervised but no significant difference between the exercise groups.

### 3.7. Comparison of Glycemic Indicators of the Study Groups at Baseline and 16 Weeks of Intervention

The supervised group exhibited significant improvements, with a reduction in HbA1C by 0.68% and FBG levels by 12.95 mg/dL. The unsupervised group showed notable improvements, with a reduction in HbA1C by 0.50% and FBG levels by 11.36 mg/dL, slightly less than the supervised group. However, the control group displayed marginal changes, with a reduction in HbA1C by 0.14% and FBG levels by 0.10 mg/dL, indicating a minimal to no effect without exercise intervention (Figure 2a,b).

These findings suggested a positive impact of the supervised exercise regimen on glycemic control. The decrease in HbA1C, which is a measure of long-term glycemic control, indicates an improvement in blood glucose regulation over the previous 2–3 months. Similarly, the reduction in FBG suggests an improved fasting glucose level, which may reflect better management of glucose levels overnight and between meals.

The observed changes underscore the potential benefits of a structured and supervised exercise program in managing key glycemic indicators among individuals in the supervised group. These data provide a foundation for discussing the role of exercise in glucose metabolism and its implications for diabetes management or prevention strategies.

### 3.8. Comparison of Lipid Profile of the Study Groups at Baseline and 16 Weeks of Intervention

The supervised group presented significant positive changes, with a reduction in triglycerides and LDL by 0.25 mmol/L and 19.70 mg/dL, respectively, and an improvement in HDL by 12.68 mg/dL. The unsupervised group exhibited positive changes but less pronounced ones than the supervised group, with a decrease in triglycerides and LDL by 0.10 mmol/L and 10.51 mg/dL, respectively, and an increase in HDL by 8.03 mg/dL. Very slight improvements were observed in the control group, with a decrease in triglycerides by 0.11 mmol/L and LDL by 2.49 mg/dL, while HDL increased by 1.48 mg/dL (Figure 2c–e).

These changes are indicative of improved lipid profiles, with a reduction in triglycerides and LDL, which are often targeted in cardiovascular risk management. The increase in HDL is associated with a reduced risk of cardiovascular events, as HDL assists in the transport of cholesterol to the liver for excretion. These data underscore the effectiveness of supervised exercise programs in positively modulating lipid profiles, which can be a critical component of managing and preventing cardiovascular diseases. These findings highlight the potential for lifestyle interventions, such as regular physical activity, to significantly impact biomarkers for lipid metabolism.

### 3.9. Comparison of Beta-Cell Function and Insulin Resistance (HOMA Indicators) of the Study Groups at Baseline and 16 Weeks of Intervention

The supervised group showed an increase in HOMA-B by 12.06, with a small increase in HOMA-IR by 0.15, which would be indicative of increased insulin resistance. The unsupervised group portrays an increase in HOMA-B by 6.00, indicating improved beta-cell function, and a small increase in HOMA-IR by 0.06. The control group demonstrates a small decrease in HOMA-B by 1.12, which could indicate reduced beta-cell function, and an increase in HOMA-IR by 0.23, suggesting increased insulin resistance (Figure 2f,g).

The increase in HOMA-B suggested an improvement in beta-cell function. This could mean that the supervised exercise regime might have increased the capacity of the pancreas to secrete insulin relative to the other groups. This is an important aspect of insulin metabolism in maintaining glucose homeostasis, and improving beta-cell function is always desirable, especially for individuals who are at risk of or are managing diabetes.

On the other hand, the rise in HOMA-IR does suggest a slightly increased insulin resistance. However, given that HOMA-IR values are all within ranges that could be considered normal variability and are generally very low, this increase is likely physiological and does not indicate a general trend.

## 4. Discussion

Global health is confronted with a challenge as a result of an estimated 700 million cases of type 2 diabetes mellitus by 2045 [8]. Pakistan, the third country in terms of diabetes prevalence [20], has reported an increase in the number of individuals living with diabetes in the past year, with an estimated 529 million individuals [47]. Prediabetes not only predicts the development of diabetes but also serves as a window of opportunity [16].

Our research examined the impact of a structured exercise manual on biochemical parameters in a prediabetic population in order to determine the efficacy and appropriateness of this intervention. We conducted a 16-week prospective, randomized controlled trial that was consistent with the American Diabetes Association’s recommendation of a minimum of 150 min of physical activity per week for prediabetes patients which is the foundation of all lifestyle interventions [18,31]. The intervention’s impact was systematically and highly comparatively quantitatively examined by dividing the study population into three groups: supervised, unsupervised, and control. It has been extensively documented that regular exercise is effective in the management of prediabetes, and lifestyle modifications have been demonstrated to reduce the incidence of diabetes by more than 50% [21]. Current study findings correspond to these advantages, as the supervised group demonstrated improvements in glycemic control and improved lipid profiles. The results are consistent with the findings of Jadhav and colleagues who discovered that supervised physical activity increased glucose utilization, as evidenced by the lower HbA1c levels of the current study participants [35]. Hemmingsen et al. concluded that exercise was essential for managing cardiovascular risk, which is supported by the reduction in LDL and triglycerides in the supervised group [7].

The comparative analysis of the efficacy of supervised and unsupervised exercise is a distinctive contribution. The results of the supervised exercise group are more significant, despite the fact that both have positive outcomes. Consequently, it is debatable whether the intervention’s efficacy is influenced by the personnel and methodology that have been trained and taught. In their evaluation of the impact of personnel on lifestyle interventions, Glechner et al. substantiated this perspective [18].

The control group’s minimal alterations serve as evidence that active intervention is indispensable for the management of prediabetes. The inverse relationship between the experimental groups and the absence of substantial initial improvements in this group underscores the critical role of exercise in preventing the progression from prediabetes to diabetes [15]. The supervised group exhibited an intriguing increase in HOMA-IR. Although initially paradoxical, it may be attributed to the adaptive response’s initial phase [5]. Consequently, this may be interpreted as a settling period, as the temporary insulin resistance is a precursor to the development of sustained improvements in insulin sensitivity. The increase in HOMA-IR in the supervised group is indeed a surprising finding. This could reflect an adaptive response to the exercise regimen (e.g., an initial increase in insulin resistance as a result of muscle repair and remodeling), but we acknowledge that further research is needed to elucidate the mechanisms behind this observation. We included a more thorough discussion of this result and its implications in the discussion section. However, the unexpected increase in HOMA-IR in the supervised group warrants further discussion. While exercise typically enhances insulin sensitivity, certain factors might lead to transient increases in HOMA-IR. Acute bouts of exercise can elevate circulating insulin levels post-exercise, potentially increasing HOMA-IR measurements if assessed shortly after exercise sessions. Additionally, individual variability in response to exercise, influenced by factors such as baseline fitness levels, diet, and genetic predispositions, can result in heterogeneous outcomes. It is also possible that the timing of post-intervention measurements captured short-term physiological responses rather than long-term adaptations. These considerations should be addressed to provide a comprehensive understanding of the results [25]. The unexpected increase in HOMA-IR underscores the complexity of metabolic adaptations to exercise in the supervised group. It may represent a temporary adaptive response rather than a negative outcome, while this finding deviates from conventional expectations. The significant improvements in beta-cell function, glycemic control, and lipid profiles suggest that the intervention remains highly beneficial. The observation of increased HOMA-IR values in the supervised group, contrary to the anticipated decrease based on the literature, presents an intriguing paradox that requires comprehensive exploration.

There are some factors that can be the cause of increased levels of HOMA-IR. Transient physiological adaptations often occurs due to acute and high-intensity exercise interventions. There can be a temporary increase in insulin resistance due to elevated insulin levels and glucose uptake in skeletal muscle, as some studies have indicated, during the immediate post-exercise phase. This phenomenon is part of the body’s short-term response to the increased metabolic demands of exercise, particularly when combining aerobic and resistance training [48]. Remodeling and repair processes in muscle tissues post-exercise, as well as transient fluctuations in insulin dynamics, also cause a temporary rise in HOMA-IR. The timing of post-intervention measurements could significantly influence HOMA-IR values. Insulin levels may have been elevated, artificially inflating HOMA-IR values, if blood samples were taken shortly after the last exercise session. To more accurately reflect baseline metabolic states, it is important to incorporate a recovery window before conducting biochemical assessments [15].

Interindividual variability may also play a role in the observed results. Cultural dietary practices and varying degrees of baseline fitness might contribute to these variations. The concurrent increase in HOMA-B values in the supervised group suggests improved beta-cell function and compensatory mechanisms. Enhanced insulin secretion from pancreatic beta cells may serve as a response to transiently increased insulin resistance, reflecting an adaptive capacity that protects against long-term glucose dysregulation. This improvement in beta-cell function is a positive outcome, as it indicates preserved pancreatic activity and a reduced risk of progression to overt diabetes anabolic responses to exercise [47]. Previous studies suggested that the supervised intervention likely involved a higher exercise intensity and stricter adherence compared to the unsupervised group. While these factors contribute to improved glycemic control and lipid profiles, they may also impose greater acute physiological demands. These demands can manifest as transient insulin resistance, which may normalize or improve over a longer period [49]. Short-term increases in HOMA-IR within the broader context of overall metabolic health improvements underscore the need for interpretation. Similar paradoxical increases have been observed in specific contexts, while most studies report a decreased HOMA-IR following exercise interventions. Acute responses to exercise, especially under supervision, may differ from chronic adaptations. The findings of this study highlight a gap in understanding the temporal dynamics of insulin sensitivity markers during and after structured exercise interventions.

The concept of individualized approaches to diabetes prevention is bolstered by the consistently large database of evidence, which reinforces the phenotypical heterogeneity of metabolic responses to lifestyle interventions. The exercise manual utilized in current research, which is pertinent to the Pakistani prediabetic cohort, has already made a significant contribution to the establishment of this conceptual framework. This may be regarded as the basis for the development of interventions in the management of diabetes and the prevention of disease from a cultural perspective [17,19].

Daniel Konig et al. conducted on lifestyle modification, in which they observed 37.7% of participants experiencing a reversal of prediabetes to normal glucose metabolism. These participants had shown a very good adherence (more than 80%) to the guided lifestyle intervention sessions. Data from the DPP have suggested that those patients, who converted to normal glucose regulation within the intervention period, subsequently experienced a significantly lower chance of developing T2DM during the 6-year follow-up [50].

Furthermore, the present research indicates that the performance differences in participant outcomes between supervised and non-supervised groups are significant, which underscores the necessity of conducting additional research on the factors that influence compliance and engagement. In reality, the patients were liberated from the restrictions that dictated the monitored training time on five days per week during the unsupervised program. However, this restriction served to enhance their attentiveness during the supervised period. The adherence issue is also evident in the findings of Bannell and colleagues, who discovered that individuals who were subjected to unsupervised exercise were unable to achieve a metabolic control that was superior to that of supervised exercise [51].

The implications of our research are substantial, particularly for resource-limited countries like Pakistan, where the healthcare system is being overwhelmed by the growing burden of diabetes. It is true that the viability of supervised exercise programs in these countries is uncertain. Nevertheless, the World Health Organization’s policy recommendation on community-based interventions for non-communicable diseases is consistent with the demonstrated benefits, which strongly support the integration of these interventions into primary healthcare [52].

Additionally, the control group’s metabolic profiles remained consistent throughout the intervention period, which was an intriguing discovery. Although this discovery may not be reassuring, it does serve as an illustration of the subtle nature of prediabetes. Therefore, the absence of intervention in this group is indicative of the fact that prediabetic individuals, who develop diabetes in the absence of active intervention, do not spontaneously return to normal metabolic function. Lifestyle modifications are the sole method of substantially altering the natural course of glucose intolerance, as is widely believed [53].

The present research presents a conundrum regarding the interaction between exercise and beta-cell function. Although the supervised group’s initial elevated HOMA-IR levels appear to respond in a manner similar to that of athletes, where increased insulin resistance is regarded as a physiological adaptation to increased muscular glucose uptake, the intervention results in an improvement in HOMA-B levels in each group of our study. The interpretation of this result is that the regimen may have a positive effect on beta-cell preservation, as previous research has supported the concept that physical exercise protects pancreatic viability [48].

The supervised intervention significantly enhanced beta-cell function in comparison to unsupervised behavior modification, which was the most significant-finding. The study highlighted distinct differences between supervised and unsupervised exercise interventions. Participants in the supervised group exhibited significantly greater improvements in glycemic control (HbA1c and FBG), lipid profiles, and insulin sensitivity markers compared to those in the unsupervised group. HbA1c levels were reduced more in the supervised group (0.68%) compared to the unsupervised group (0.50%). Triglycerides, LDL, and HDL changes were more pronounced in the supervised group, indicating better cardiovascular health outcomes. These differences suggest that supervised interventions offer greater efficacy in achieving metabolic health improvements. There are some behavior factors that should be addressed. Supervised programs provide external monitoring that show accountability, which likely increases participant adherence and consistency in following prescribed routines. Access to trained personnel ensures that exercises are performed correctly and at the right intensity, maximizing benefits. The presence of a supervisor creates a structured environment that fosters motivation and reduces feelings of isolation, common in unsupervised programs. Participants may feel more supported and encouraged when engaging with supervisors or peers in the program. Supervised groups often report higher adherence due to regular follow-ups and progress tracking, which are lacking in unsupervised settings where participants are left to self-regulate. There are some challenges that must be faced while supervising groups. Supervised programs require trained personnel, facilities, and monitoring tools, which may be difficult to sustain in low-resource settings. Ensuring consistent access to facilities and resources for large populations can be challenging. In some regions, societal norms or work schedules may hinder participation in structured programs.

### Limitations

The 16 Weeks duration limits the ability to observe long-term effects, particularly on sustained behavioral changes or the prevention of diabetes progression. Although it is important to note some limitations of current study, specifically the subjectivity of the self-reporting in the non-supervised group, which may introduce a level of bias that could account for inaccuracies in the interpretation of the detriment. The study was conducted in a single urban center, limiting the generalizability of the findings to rural or diverse populations. The upper age limit of 40 is more prone to prediabetes. This was a limitation in the current study; only an 18 to 40 years age range was included. In any event, our research is a stepping stone to future studies.

## 5. Strength of the Current Research Work

The reliability of these findings is enhanced by the randomized design of the study, which minimizes selection bias and guarantees that the groups are comparable.

The structured exercise program, which emphasizes both aerobic and resistance components, offers a well-defined approach to enhancing the glycemic and lipid profiles of prediabetic patients. The intervention’s metabolic impact was treated to a comprehensive comprehension through the measurement of multiple biochemical parameters, including FBG, HbA1c, lipid profiles, HOMA-B, and HOMA-IR. The study’s comparison of supervised and unsupervised exercise groups provides valuable insights into the significance of supervision in exercise interventions. The findings are pertinent to populations at a high risk for diabetes, as they address a significant public health issue in a resource-limited context by concentrating on prediabetic individuals from Pakistan.

### Future Directions

Increase the sample size and include participants from diverse geographic and socioeconomic backgrounds to enhance its generalizability. Include the comprehensive tracking of lifestyle factors beyond exercise, such as diet, sleep, and stress management, to better understand their interaction with the intervention outcomes. Use longitudinal study designs to evaluate the long-term effects of structured exercise interventions on biochemical markers and the progression from prediabetes to type 2 diabetes. Extended follow-up periods will allow for a deeper understanding of the sustainability of metabolic improvements, the durability of adherence to lifestyle interventions, and their potential to mitigate the risk of diabetes-related complications over time. Use objective adherence measures, such as wearable fitness trackers, to monitor activity levels and compliance. Conduct regular follow-ups with the unsupervised group to encourage accountability. A combination of supervised sessions and technology-driven self-monitoring (e.g., mobile apps or wearable devices) could offer a middle ground, balancing cost and efficacy. Community-based supervision models can leverage local health workers to provide basic monitoring.

In general, our research contributes to the ongoing corpus of evidence and provides a comprehensive perspective on the intricate prediabetes landscape. The intervention in question facilitated the discovery of the double action of exercise intervention, which, in the context of the exercise manual, could provide a viable approach to reducing the prevalence of prediabetes in Pakistan. However, the practical implications of scaling these methods will be difficult to resolve, necessitating the collaboration of policymakers and public health officials, as well as the support of community.

## 6. Conclusions

In conclusion, it is necessary to mention the urgent need for public health practices aimed at preventing T2DM based on lifestyle modifications. Professionals and clinicians should use the results of the present study and provide evidence-based interventions that are sustainable and feasible. This exercise manual is beneficial in healthcare settings as well at community level and provide standard exercise guidelines to control prediabetes progression into diabetes mellitus. This exercise manual program can help to improve glucose levels in prediabetic populations.

## Figures and Tables

**Figure 1 medicina-61-00190-f001:**
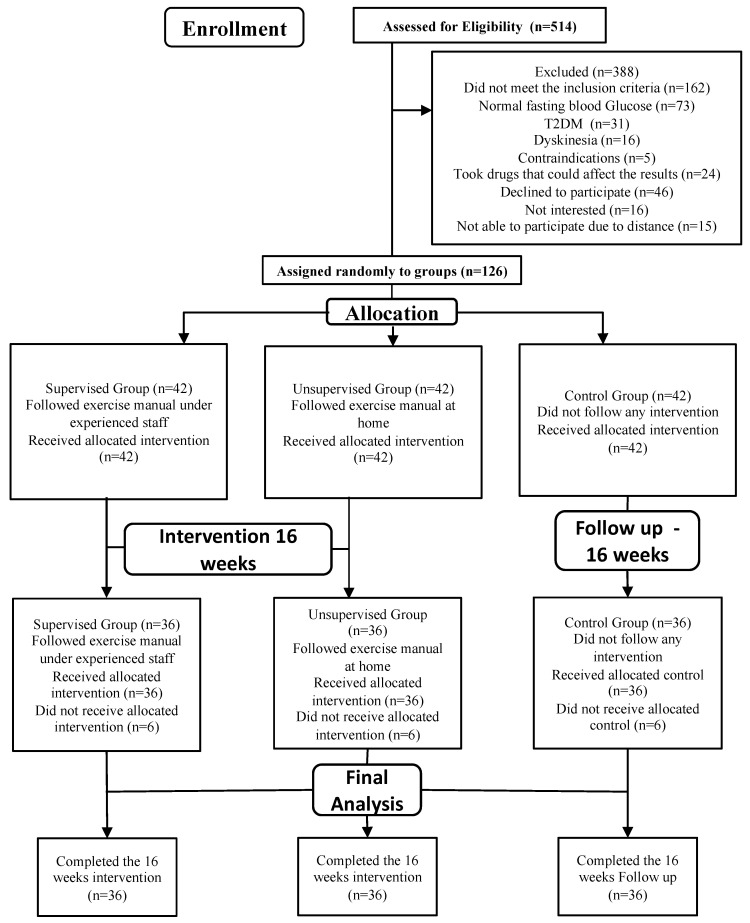
Consort flow diagram of the research methodology.

**Figure 2 medicina-61-00190-f002:**
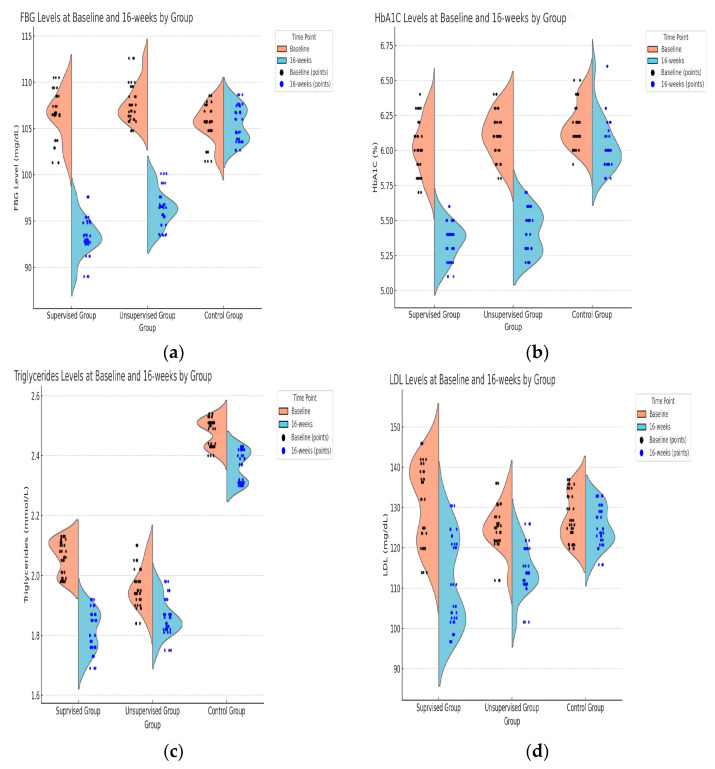
Comparison of glycemic indicators of study groups at baseline and 16 Weeks of intervention: (**a**) FBG, (**b**) HbA1C, lipid profile, (**c**) TG, (**d**) LDL, (**e**) HDL and insulin function and resistance, (**f**) HOMA-B, and (**g**) HOMA-IR.

**Table 1 medicina-61-00190-t001:** Clinical reference range of all biochemical variables of study.

Biochemical Variable	Reference Range	Interpretation
Fasting Blood Glucose (FBG)	70–99 mg/dL	Normal fasting glucose level
	100–125 mg/dL	Prediabetes (impaired fasting glucose)
	≥126 mg/dL	Diabetes mellitus (requires confirmation)
HbA1c	<5.7%	Normal glycemic control
	5.7–6.4%	Prediabetes
	≥6.5%	Diabetes mellitus (requires confirmation)
Triglyceride	<150 mg/dL	Normal
	150–199 mg/dL	Borderline high
	200–499 mg/dL	High
	≥500 mg/dL	Very high
Low-Density Lipoprotein (LDL)	<100 mg/dL	Optimal
	100–129 mg/dL	Near optimal
	130–159 mg/dL	Borderline high
	160–189 mg/dL	High
	≥190 mg/dL	Very high
High-Density Lipoprotein (HDL)	>40 mg/dL (men), >50 mg/dL (women)	Recommended level
	<40 mg/dL (men), <50 mg/dL (women)	Low, associated with increased cardiovascular risk
HOMA-B (Beta-Cell Function)	100% (varies by population)	Indicates pancreatic beta-cell function; no strict “normal” range as it varies with study population
HOMA-IR (Insulin Resistance)	<1.0	Optimal insulin sensitivity
	1.0–1.9	Indicates mild insulin resistance
	2.0–2.9	Moderate insulin resistance
	≥3.0	Significant insulin resistance

**Table 2 medicina-61-00190-t002:** Demographic characteristics and biochemical variables of experimental and control groups [mean ± SD] at the baseline.

Variables		Supervised Group (*n* = 36) Mean ± SD, %	Unsupervised Group (*n* = 36) Mean ± SD, %	Control Group (*n* = 36) Mean ± SD, %	F/χ^2^	*p*
Gender	M	15 (41.7%)	20 (56.6%)	19 (52.8%)	1.7	>0.05 ^b^
	F	21 (58.3%)	16 (44.4%)	17 (47.2%)	1.3	>0.05 ^b^
Age (years)		32.25 ± 5.69	30.58 ± 3.30	30.75 ± 4.14	1.50	>0.05 ^a^
Weight (kg)		85.03 ± 4.7	81.08 ± 3.95	86.83 ± 4.33	16.54	<0.05
Height (cm)		166.16 ± 3.60	168.58 ± 4.42	170.08 ± 4.43	8.07	<0.05 ^a^
BMI (kg/m^2^)		30.58 ± 1.25	28.60 ± 1.58	30.58 ± 6.02	14.50	<0.05 ^a^
FBG (mg/dL)		106.35 ± 2.56	107.68 ± 2.11	105.6 ± 2.02	7.9	<0.05 ^a^
HbA1C (%)		6.02 ± 0.19	6.11 ± 0.16	6.18 ± 0.16	7.32	<0.05 ^a^
TGs (mmol/L)		2.06 ± 0.05	1.95 ± 0.07	2.47 ± 0.04	786.3	<0.05 ^a^
LDL (mg/dL)		131.20 ± 10.15	125.11 ± 5.90	127.65 ± 5.98	5.8	<0.05 ^a^
HDL (mg/dL)		36.81 ± 3.32	36.33 ± 2.36	39.98 ± 2.51	18.55	>0.05 ^a^
HOMA-B		69.03 ± 1.20	71.46 ± 2.31	79.28 ± 3.55	18.46	<0.05 ^a^
HOMA-IR		1.00 ± 0.04	0.99 ± 0.02	1.03 ± 0.02	494.02	>0.05 ^a^

^a^ *p* for numerical variables; ^b^
*p* for categorial variables. FBG, fasting blood glucose; HbA1c, hemoglobin A1c; TGs, triglycerides; LDL, low-density lipoprotein; HDL, high-density lipoprotein; HOMA-IR, homeostasis model assessment of insulin resistance; and HOMA-B, homeostasis model assessment of beta-cell function.

**Table 3 medicina-61-00190-t003:** Effects of interventions on biochemical indicators in group A (who followed exercise manual under supervision).

Variables	Baseline (M ± SD)	16 Weeks (M ± SD)	t (35), *p*	Cohen’s d
FBG	106.26 ± 2.59	93.44 ± 2.17	38.41, *p* < 0.001	5.35
HbA1C	6.02 ± 0.19	5.34 ± 0.14	25.31, *p* < 0.001	0.71
TGs	2.06 ± 0.06	1.81 ± 0.07	41.48, *p* < 0.001	3.95
LDL	131.32 ± 10.18	112.00 ± 1.96	19.64, *p* < 0.001	2.65
HDL	36.88 ± 0.57	49.48 ± 0.64	−22.46, *p* < 0.001	21.56
HOMA-B	69.03 ± 1.24	81.10 ± 2.23	−11.23, *p* < 0.001	6.70
HOMA-IR	1.01 ± 0.04	1.16 ± 0.02	−31.96, *p* < 0.001	4.63

*p* values calculated with paired *t* test. FBG, fasting blood glucose; HbA1c, hemoglobin A1c; TGs, triglycerides; LDL, low-density lipoprotein; HDL, high-density lipoprotein; HOMA-IR, homeostasis model assessment of insulin resistance; and HOMA-B, homeostasis model assessment of beta-cell function.

**Table 4 medicina-61-00190-t004:** Effects of interventions on biochemical indicators in group B (exercise manual was followed at home after initial guidelines).

Variables	Baseline (M ± SD)	16 Weeks (M ± SD)	t (34), *p*	Cohen’s d
FBG	107.68 ± 2.11	96.32 ± 1.94	42.04, *p* < 0.001	5.60
HbA1C	6.11 ± 0.17	5.42 ± 0.16	24.96, *p* < 0.001	0.41
Triglyceride	1.95 ± 0.70	1.86 ± 0.06	31.77, *p* < 0.001	0.18
LDL	125.11 ± 5.90	114.60 ± 6.28	56.97, *p* < 0.001	1.72
HDL	36.33 ± 2.37	44.35 ± 4.25	−9.42, *p* < 0.001	2.33
HOMA-B	71.46 ± 2.31	77.46 ± 1.83	−26.84, *p* < 0.001	2.87
HOMA-IR	0.99 ± 0.02	1.05 ± 0.04	−15.80, *p* < 0.001	1.94

*p* values calculateed with paired *t* test. FBG, fasting blood glucose; HbA1c, hemoglobin A1c; TGs triglyceride; LDL, low-density lipoprotein; HDL, high-density lipoprotein; HOMA-IR, homeostasis model assessment of insulin resistance; and HOMA-B, homeostasis model assessment of beta-cell function.

**Table 5 medicina-61-00190-t005:** Effects of interventions on biochemical indicators in group C (control group).

Variables	Baseline (Mean ± SD)	16 Weeks (Mean ± SD)	t (32), *p*	Cohen’s d
FBG	105.60 ± 2.02	105.50 ± 1.88	0.392, >0.05	0.00
HbA1C	6.18 ± 0.16	6.04 ± 0.195	8.171, >0.05	0.09
Triglyceride	2.47 ± 0.045	2.36 ± 0.053	48.062, >0.05	2.23
LDL	127.65 ± 5.98	125.17 ± 5.26	56.974, >0.05	0.44
HDL	39.98 ± 2.51	41.47 ± 3.37	−4.437, >0.05	0.50
HOMA-B	79.28 ± 3.55	78.16 ± 2.39	3.602, >0.05	2.67
HOMA-IR	1.034 ± 0.023	1.262 ± 0.024	−46.11, <0.001	9.62

*p* values calculated with paired *t* test. FBG, fasting blood glucose; HbA1c, hemoglobin A1c; TGs, triglyceride; LDL, low-density lipoprotein; HDL, high-density lipoprotein; HOMA-IR, homeostasis model assessment of insulin resistance; and HOMA-B, homeostasis model assessment of beta-cell function.

**Table 6 medicina-61-00190-t006:** Biochemical variables between three prediabetic groups.

Biochemical Variables	Group	Before Mean (SD)	After Mean (SD)	*p*-Value *	Cohen’s d	95% CI Lower	95% CI Upper
HbA1C	Supervised	6.02 (0.19)	5.35 (0.13)	0.00	−4.44	−5.54	−3.35
	Unsupervised	6.12 (0.17)	5.43 (0.16)	0.00	−4.16	−5.19	−3.13
	Control	6.18 (0.16)	6.04 (0.20)	0.00	−1.36	−1.82	−0.91
FBG	Supervised	106.36 (2.57)	93.41 (2.12)	0.00	−6.57	−8.14	−5.00
	Unsupervised	107.69 (2.11)	96.32 (1.95)	0.00	−7.01	−8.68	−5.33
	Control	105.61 (2.03)	105.51 (1.89)	0.70	−0.07	−0.39	0.26
HOMA-B	Supervised	69.03 (1.21)	81.09 (2.17)	0.00	5.26	3.98	6.53
	Unsupervised	71.47 (2.32)	77.47 (1.84)	0.00	4.47	3.38	5.57
	Control	79.28 (3.56)	78.16 (2.40)	0.00	−0.60	−0.96	−0.24
HOMA-IR	Supervised	1.01 (0.04)	1.16 (0.02)	0.00	5.33	4.04	6.62
	Unsupervised	0.99 (0.02)	1.05 (0.04)	0.00	2.63	1.93	3.33
	Control	1.03 (0.02)	1.26 (0.02)	0.00	7.68	5.85	9.51
Triglyceride	Supervised	2.06 (0.06)	1.81 (0.07)	0.00	−7.31	−9.06	−5.57
	Unsupervised	1.96 (0.07)	1.86 (0.06)	0.00	−5.30	−6.58	−4.01
	Control	2.48 (0.05)	2.37 (0.05)	0.00	−8.01	−9.92	−6.11
LDL	Supervised	131.21 (10.16)	111.51 (11.26)	0.00	−3.14	−3.94	−2.33
	Unsupervised	125.12 (5.91)	114.60 (6.29)	0.00	−9.50	−11.74	−7.25
	Control	127.65 (5.98)	125.16 (5.26)	0.00	−1.01	−1.42	−0.61
HDL	Supervised	36.82 (3.32)	49.50 (3.74)	0.00	3.97	2.98	4.95
	Unsupervised	36.33 (2.37)	44.36 (4.25)	0.00	1.57	1.08	2.06
	Control	39.99 (2.52)	41.47 (3.37)	0.00	0.74	0.37	1.11

* *p*-value for group comparison derived from GLMM allowing for clustering and adjusted using ITT analysis based on MI.

**Table 7 medicina-61-00190-t007:** Biochemical variable analysis between the groups—Mixed-Effect Model analysis.

Variable	Effect	Estimate	Std Error	SS	*p*-Value	Significance
HbA1C	Baseline Mean	6.181	0.028	38.19	0.00	***
	Time Effect	−0.138	0.039	0.019	0.00	***
	Unsupervised vs. Supervised	−0.064	0.039	0.004	0.10	
	Time × Unsupervised Interaction	−0.554	0.056	0.30	0.00	***
FBG	Baseline Mean	105.605	0.349	11,152.41	0.00	***
	Time Effect	−0.098	0.493	0.01	0.84	
	Unsupervised vs. Supervised	2.081	0.493	4.33	0.00	***
	Time × Unsupervised Interaction	−11.267	0.697	126.93	0.00	***
HOMA-B	Baseline Mean	79.281	0.387	6285.45	0.00	***
	Time Effect	−1.118	0.548	1.25	0.04	*
	Unsupervised vs. Supervised	−7.813	0.548	61.04	0.00	***
	Time × Unsupervised Interaction	7.116	0.775	50.63	0.00	***
HOMA-IR	Baseline Mean	1.034	0.005	1.07	0.00	***
	Time Effect	0.227	0.007	0.05	0.00	***
	Unsupervised vs. Supervised	−0.043	0.007	0.002	0.00	***
	Time × Unsupervised Interaction	−0.168	0.01	0.02	0.00	***
Triglyceride	Baseline Mean	2.477	0.01	6.13	0.00	***
	Time Effect	−0.11	0.014	0.01	0.00	***
	Unsupervised vs. Supervised	−0.518	0.014	0.26	0.00	***
	Time × Unsupervised Interaction	0.011	0.02	0	0.59	
LDL	Baseline Mean	127.651	1.287	16,294.73	0.00	***
	Time Effect	−2.486	1.821	6.179	0.17	
	Unsupervised vs. Supervised	−2.534	1.82	6.422	0.16	
	Time × Unsupervised Interaction	−8.028	2.575	64.45	0.00	**
HDL	Baseline Mean	39.989	0.547	1599.13	0.00	***
	Time Effect	1.483	0.774	2.2	0.06	
	Unsupervised vs. Supervised	−3.659	0.774	13.39	0.00	***
	Time × Unsupervised Interaction	6.542	1.094	42.80	0.00	***

(***) *p* value less than 0.001, (**) *p* value less than 0.01, (*) *p* value less than 0.05. *p* values calculated with Mixed-Effect Model analysis. FBG, fasting blood glucose; HbA1c, hemoglobin A1c; TG, triglycerides; LDL, low-density lipoprotein; HDL, high-density lipoprotein; HOMA-IR, homeostasis model assessment of insulin resistance; and HOMA-B, homeostasis model assessment of beta-cell function.

## Data Availability

Available on request from the corresponding author.

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
