# Peer review of "Impact of Exercise Manual Program on Biochemical Markers in Sedentary Prediabetic Patients: A Randomized Controlled Trial"

_medicina, 2025, doi:10.3390/medicina61020190_

Round 1
Reviewer 1 Report
Comments and Suggestions for Authors
The manuscript presents a promising approach and offers valuable insights into prediabetes management. However, revisions are required to address certain limitations :
Introduction
- The manuscript does not explicitly identify gaps in the existing literature regarding supervised versus unsupervised exercise interventions. A clear discussion of these gaps would strengthen the study's rationale.
- While the high prevalence of prediabetes in Pakistan is highlighted, the manuscript does not explore culturally specific barriers or facilitators to exercise adherence. This aspect is particularly relevant, as cultural factors can significantly influence motivation and participation in both supervised and unsupervised interventions.
- Expanding on the biological mechanisms through which exercise impacts biochemical markers such as HbA1c, lipid profiles, and HOMA indices would strengthen the link between the intervention and the outcomes reported.
Methods
- Important details necessary for study replication are missing, such as the specific intensity levels for aerobic and resistance exercises (e.g., percentage of maximum heart rate).
- The criteria for monitoring adherence in the unsupervised group are insufficiently explained, potentially introducing bias in adherence data.
- The upper age limit of 40 years may exclude older individuals who are more representative of prediabetic populations. This should be acknowledged as a limitation.
- The precision of adherence measurement, particularly in the unsupervised group, appears unclear and could introduce variability in the results.
- While the statistical methods appear appropriate, no adjustments for multiple comparisons, such as Bonferroni correction, are mentioned. Addressing this would reduce the risk of Type I errors.
Results
- Self-reported adherence in the unsupervised group may introduce bias, possibly overstating the effectiveness of this intervention.
- The significant improvements in glycemic control (HbA1c, fasting blood glucose) and lipid profiles (HDL, LDL, triglycerides) observed in the supervised group align with the study's conclusions. However, the increase in HOMA-IR in the supervised group is not adequately explained. While this could reflect an adaptive physiological response, its absence in the discussion raises concerns.
Discussion
- The conclusion that the exercise manual could broadly reduce prediabetes prevalence may be premature without longer-term follow-up or replication in diverse populations.
- The control group did not receive a placebo or active intervention, which could have introduced a Hawthorne effect. Additionally, the study missed an opportunity to compare other lifestyle modifications, such as dietary counseling. Addressing this issue would strengthen the discussion.
- While some limitations, such as the reliance on self-reported adherence, are acknowledged, others are insufficiently addressed. These include the homogeneity of the sample and the short duration of the study. Additionally, the unexpected increase in HOMA-IR in the supervised group remains unexplained, leaving a critical result open to interpretation
Author Response
Response to Reviewers (01)Comments
INTRODUCTION
1.The manuscript does not explicitly identify gaps in the existing literature regarding supervised versus unsupervised exercise interventions. A clear discussion of these gaps would strengthen the study's rationale.
ANS: Thank you for your suggestion. In literature there are studies that compare the supervised with unsupervised groups, supervised with control group. But there is a gap in literature about comparison of supervised, unsupervised and control group. The framework of exercise manual based on content that can be easily performed with no hassles and cost effectiveness. As our cultural norms showed hindrance for outdoor exercise. Purpose to add unsupervised group to motivate the participants in performing exercises on their own. It showed some novelty in our work to add comparison of supervised, unsupervised and control groups. This has been added in Introduction
- While the high prevalence of prediabetes in Pakistan is highlighted, the manuscript does not explore culturally specific barriers or facilitators to exercise adherence. This aspect is particularly relevant, as cultural factors can significantly influence motivation and participation in both supervised and unsupervised interventions.
Ans: We appreciate the reviewer’s suggestion to discuss culturally specific barriers and facilitators to exercise adherence in the Pakistani context. This content added in introduction. In the revised manuscript, we have included a section in the introduction that highlights the following:
Cultural Barriers:
Gender roles and societal norms in Pakistan may restrict outdoor physical activity, particularly for women, due to concerns about safety and privacy.
Limited availability of female-only fitness facilities and a lack of culturally acceptable exercise attire. Family responsibilities often prioritized over individual health, reducing time availability for structured exercise. This exercise manual provides the structured exercise program that can be easily performed indoor with no equipment’s need and also its cost effective.
Facilitators:
Community-based programs and group exercises have shown promise in overcoming barriers by creating a sense of collective motivation. This exercise manual Programme is about community awareness and prevention to diabetes. Increasing awareness through culturally tailored health education campaigns can also serve as a facilitator.
- Expanding on the biological mechanisms through which exercise impacts biochemical markers such as HbA1c, lipid profiles, and HOMA indices would strengthen the link between the intervention and the outcomes reported.
Ans: The selected biochemical markers HbA1c, fasting blood glucose (FBG), lipid profiles and insulin sensitivity indicators are integral to understanding the metabolic and physiological impacts of structured exercise interventions in prediabetic patients. Each marker aligns with the study objectives as follows:
Glycemic Control: HbA1c and Fasting Blood Glucose (FBG)
HbA1c reflects long-term glycemic control over the preceding 2–3 months, making it a critical indicator for evaluating sustained changes in glucose metabolism due to the interventions. FBG provides immediate and daily snapshots of blood glucose regulation, helping to assess short-term metabolic responses. These markers are essential for tracking the progression or reversal of prediabetes, as they are diagnostic criteria for both prediabetes and diabetes.
Lipid Profile: Triglycerides, LDL, and HDL
Triglycerides and LDL (low-density lipoprotein) are key markers of cardiovascular risk, often elevated in prediabetic and diabetic conditions due to dysregulated lipid metabolism. HDL (high-density lipoprotein), the “good cholesterol,” is inversely associated with cardiovascular risk. Improvements in HDL are indicative of better lipid clearance and metabolic health. Including these markers aligns with the study's goal of understanding cardiovascular risk modulation through exercise.
Insulin Sensitivity and Beta-cell Function: HOMA-IR and HOMA-B
HOMA-IR (Homeostatic Model Assessment of Insulin Resistance) quantifies insulin resistance, a central feature of prediabetes. HOMA-B assesses pancreatic beta-cell function, crucial for understanding how exercise interventions preserve or improve insulin secretion capacity. These markers allow the evaluation of how the interventions address the underlying pathophysiology of prediabetes, such as insulin resistance and beta-cell dysfunction.
METHODS:
- 4. Important details necessary for study replication are missing, such as the specific intensity levels for aerobic and resistance exercises (e.g., percentage of maximum heart rate)
Participants performed moderate intensity with compliance of 70-75% HR peak separated by 40 seconds of easy recovery time.
- The criteria for monitoring adherence in the unsupervised group are insufficiently explained, potentially introducing bias in adherence data.
Thank you for your valuable suggestion. Unsupervised group provided same exercise manual with same dietary instructions. Participants recorded their exercise routine in a logbook during home training days. Participants were asked to record (a) the activity performed, (b) the minutes spent engaging in the activity, (c) Frequency of session and (d) how hard the session. Results showed less changes in biochemical markers as compared to supervised group that lessen the biasness in adherence data.
- The upper age limit of 40 years may exclude older individuals who are more representative of prediabetic populations. This should be acknowledged as a limitation
Thank you for noting this detail. Upper age limit of 40 years has been added in limitations.
- The precision of adherence measurement, particularly in the unsupervised group, appears unclear and could introduce variability in the results.
Thank you for the suggestion. Statistically results showed supervised group more effective as compared to unsupervised group. This may reduce the effect of biasness.
- While the statistical methods appear appropriate, no adjustments for multiple comparisons, such as Bonferroni correction, are mentioned. Addressing this would reduce the risk of Type I errors.
I appreciate your insightful comment about the need for adjustments for multiple comparisons. We acknowledge that multiple comparison corrections, such as the Bonferroni correction, are valuable for controlling the risk of Type I errors when multiple hypotheses are tested simultaneously.
In this study, we chose not to apply such corrections as the primary focus was on identifying key trends and associations, and the number of comparisons was relatively small. Additionally, we believe that the statistical methods used are robust enough to provide reliable insights within the scope of this analysis. However, we agree that for studies involving a larger number of comparisons or in situations where the risk of Type I errors may be heightened, applying corrections such as Bonferroni or False Discovery Rate (FDR) would be prudent.
We will consider this adjustment in future analyses to ensure the results are as conservative as possible and to further reduce the potential for Type I errors.
Thank you again for your valuable suggestion.
RESULTS:
- Self-reported adherence in the unsupervised group may introduce bias, possibly overstating the effectiveness of this intervention.
Thank you for this suggestion. As results showed that supervised group showed more improvement in biochemical changes as compared to unsupervised to lessen the biasness of intervention. Main purpose of unsupervised group as it helps to decrease the reliance on trainer and encourage the participant to do exercise on their own as to overcome cultural barrier and cost effectiveness. In future studies some tracker monitoring device can be suggest in unsupervised group to reduce the biasness in intervention.
- The significant improvements in glycemic control (HbA1c, fasting blood glucose) and lipid profiles (HDL, LDL, triglycerides) observed in the supervised group align with the study's conclusions. However, the increase in HOMA-IR in the supervised group is not adequately explained. While this could reflect an adaptive physiological response, its absence in the discussion raises concerns.
The increase in HOMA-IR in the supervised group is indeed a surprising finding. This could reflect an adaptive response to the exercise regimen (e.g., an initial increase in insulin resistance as a result of muscle repair and remodeling), but we acknowledge that further research is needed to elucidate the mechanisms behind this observation. We included a more thorough discussion of this result and its implications in the discussion section. However, the unexpected increase in HOMA-IR in the supervised group warrants further discussion. While exercise typically enhances insulin sensitivity, certain factors might lead to transient increases in HOMA-IR. Acute bouts of exercise can elevate circulating insulin levels post-exercise, potentially increasing HOMA-IR measurements if assessed shortly after exercise sessions. Additionally, individual variability in response to exercise, influenced by factors such as baseline fitness levels, diet, and genetic predispositions, can result in heterogeneous outcomes. It's also possible that the timing of post-intervention measurements captured short-term physiological responses rather than long-term adaptations. These considerations should be addressed to provide a comprehensive understanding of the results.
DISCUSSION:
- The conclusion that the exercise manual could broadly reduce prediabetes prevalence may be premature without longer-term follow-up or replication in diverse populations.
Thank you for this suggestion. Exercise manual helps in improving biochemical variables. But Long term follow up have added in recommendations.
- The control group did not receive a placebo or active intervention, which could have introduced a Hawthorne effect.
Control group provided the log book in which they mentioned their daily routine. As we have to observe exercise effects in supervised and unsupervised groups so physical activity couldn’t mention for control group.
- The study missed an opportunity to compare other lifestyle modifications, such as dietary counseling. Addressing this issue would strengthen the discussion.
Thank you for mentioning this point. In discussion we added one more study relevant to lifestyle modification.
- While some limitations, such as the reliance on self-reported adherence, are acknowledged, others are insufficiently addressed. These include the homogeneity of the sample and the short duration of the study. Additionally, the unexpected increase in HOMA-IR in the supervised group remains unexplained, leaving a critical result open to interpretation
Thank you for your concern. Limitations have been improved as per your guidance.

Reviewer 2 Report
Comments and Suggestions for Authors
Dear Authors,
Thank you for your exciting work on the impact of structured exercise programs on biochemical markers in sedentary prediabetic patients. Your study addresses a significant public health concern. It provides valuable insights into the role of supervised and unsupervised exercise interventions in preventing the progression of prediabetes to Type 2 diabetes mellitus (T2DM). Below, I provide detailed comments and suggestions to enhance the clarity, depth, and rigour of your manuscript.
Abstract
The abstract is clear and provides a concise summary of the study. No significant changes are required. It effectively outlines the background, methodology, and critical findings.
Introduction
Your introduction effectively sets the stage for the study. However, the discussion on the role of physical activity (PA) in T2DM prevention could be expanded:
- Literature Expansion: The role of PA (line 84) should be enhanced by analysing recent literature addressing what types of PA are most effective. Please include findings from the following references:
Ø Fanelli et al., 2022, highlight the benefits of structured aerobic and resistance training programs in improving adolescent metabolic outcomes (Fanelli E, Abate Daga F, Pappaccogli M, Eula E, Astarita A, Mingrone G, Fasano C, Magnino C, Schiavone D, Rabbone I, Gollin M, Rabbia F, Veglio F. A structured physical activity program in an adolescent population with overweight or obesity: a prospective interventional study. Appl Physiol Nutr Metab. 2022 Mar;47(3):253-260. doi: 10.1139/apnm-2021-0092. Epub 2021 Oct 27. PMID: 34706211.)
Ø Schranz et al., 2014, detail the effects of resistance training on strength, body composition, and psychological health in overweight adolescents. (Schranz N, Tomkinson G, Parletta N, Petkov J, Olds T. Can resistance training change the strength, body composition and self-concept of overweight and obese adolescent males? A randomised controlled trial. Br J Sports Med. 2014 Oct;48(20):1482-8. doi: 10.1136/bjsports-2013-092209. Epub 2013 Aug 14. PMID: 23945035.)
- Specific Suggestions: Answer the question, "What type of PA works best?" and discuss:
Ø The comparative effectiveness of aerobic versus resistance training.
Ø The potential benefits of combined programs on insulin sensitivity and lipid profiles.
- Implications: Link these findings to preventing prediabetes progression and managing metabolic health.
Methods
This section is detailed but requires some clarifications and improvements:
- Formatting:
Ø Ensure consistent formatting at lines 139 and 146.
- Participant Selection:
Ø Specify that participants must be “sedentary” to avoid BMI variations due to regular physical activity.
- Randomization and Group Details:
Ø Clarify the randomization process (e.g., simple, block, or stratified randomisation).
Ø Describe the activities of Group C in detail (e.g., whether they received medical therapy, dietary advice, or only measurements).
Ø Provide a breakdown of the gender distribution for each group.
- Intervention Description:
Ø Rewrite lines 163–200 for clarity. Give a linear and logical order to all the information you declared. In addition, a structured table summarising each group's exercises, sets, repetitions, and recovery times would make this section easier to follow.
Ø Include precise progression details for aerobic and resistance training components.
- Outcome Measures:
Ø Justify the choice of biochemical markers and their relevance to the study objectives.
Results
The results section is comprehensive and statistically robust. No major revisions are needed, and the tables and figures effectively communicate the findings.
Discussion
The discussion is well-written and contextualises the findings appropriately. However, I suggest:
- Supervised vs. Unsupervised:
Ø Expand on the differences observed between supervised and unsupervised interventions. Discuss why supervised programs might have shown greater efficacy and include behavioural or psychological factors that could have influenced adherence.
- Implications of Findings:
Ø Highlight how the findings could guide future interventions in resource-limited settings.
Ø Discuss the potential scalability of supervised exercise programs and the challenges associated with implementation.
- Limitations:
Ø Address the study's limitations, particularly the short duration of the intervention and potential biases in self-reported adherence in the unsupervised group.
Ø Suggest how these limitations could be addressed in future studies.
Conclusion
The conclusion aligns well with the study's objectives and findings. It effectively emphasises the importance of structured exercise programs in managing prediabetes. No changes are necessary here.
Additional Suggestions
- Compliance and Adherence:
Ø Discuss strategies to improve adherence in unsupervised programs, potentially drawing on behavioural research or using digital tools for monitoring.
- Figures and Tables:
Ø Ensure all figures and tables are labelled clearly and referenced appropriately in the text.
- Future Directions:
Ø Longitudinal studies are recommended to evaluate the long-term effects of interventions on biochemical markers and the progression of diabetes.
Author Response
Response to Reviewer (02) Comments
INTRODUCTION:
1.The discussion on the role of physical activity (PA) in T2DM prevention could be expanded:
Thank you for your valuable suggestion. Structured Physical activity involves all body movement that increases energy use while performing exercise. To prevent or delay type 2 diabetes mellitus exercise plays important role as it lowers blood glucose level, lessen the cardiovascular risk factors, important part in weight loss and improves well-being. Diabetes type, activity type, and presence of diabetes-related complications are varied with the challenges related to blood glucose management. Physical activity and exercise recommendations, therefore, should be tailored to meet the specific needs of each individual.
Fourth risk factor for global mortality is Physical inactivity and is estimated to be the main cause of 27% of diabetes cases. Physical activity can be mild, moderate or intense; the higher the performance, the greater the results showed. On the other hand, it is convenient to point out that regardless of the PA performed, spending a lot of time on sedentary behaviors can additionally increase health risks.
- Literature Expansion: The role of PA (line 84) should be enhanced by analyzing recent literature addressing what types of PA are most effective. Please include findings from the following references:
Thank you for your kind suggestions. We have added the relevant content.
3 "What type of PA works best?" and discuss: The comparative effectiveness of aerobic versus resistance training.
Thank you for positive feedback. We added content in manuscript.
The most commonly used regimes of exercise therapy are aerobic exercise and resistance training. Its until in discussion that which training is most effective. Both exercise regimes are best and having their own benefits on muscles and glucose monitoring. Studies have shown that aerobic exercises can stabilize blood pressure, enhance lipoprotein density, and enhance cardiovascular function, thus decreasing the risk of cardiovascular disease. On the other hand, Resistance Training is primarily beneficial as it improves the function and mass of skeletal muscles, as well as energy expenditure and metabolic activity in muscle tissue. In some studies, combined training has been shown to improve lipids and vascular function more effectively than AE or RT alone Evidence suggests that Aerobic exercise and resistance training have beneficial effects on various participants.
- The potential benefits of combined programs on insulin sensitivity and lipid profiles.
I appreciate your attention into this detail. Both aerobic and resistance exercise within a single session had a positive impact on body mass and composition as well as enhanced glucose and lipid metabolism which includes. Benefits of combined training in improving glycemic management and lipid parameter health in obese adults are proven by many research studies. Current ADA guidelines advocate a combination of aerobic and resistance training, which may be the most beneficial exercise modality for regulating lipid profile and glucose levels.
- Implications: Link these findings to preventing prediabetes progression and managing metabolic health.
Thank you for your kind suggestion. Regular exercise training reduces A1C, triglycerides, blood pressure, and insulin resistance in individuals with prediabetes. Alternatively, Resistance training promotes rapid enhancement of skeletal muscle oxidative capacity, insulin sensitivity, and glycemic control in adults with prediabetes. For low muscular strength and accelerated decline in muscle strength and functional status diabetes is an independent risk factor .The health benefits of resistance training for all adults include improvements in muscle mass, body composition, strength, physical function, mental health, bone mineral density, insulin sensitivity, blood pressure, lipid profiles, and cardiovascular health .Performing resistance exercise first results in less hypoglycemia than when aerobic exercise is performed when resistance and aerobic exercise are undertaken in one exercise session. For individuals with type 2 diabetes combination of aerobic and resistance training benefits include improvements in glycemic control, insulin resistance, fat mass, blood pressure, strength, and lean body mass.
METHODS:
- Formatting:
Ø Ensure consistent formatting at lines 139 and 146.
Thank you for feedback. I have been formatted these lines in manuscript.
- Participant Selection:
Ø Specify that participants must be “sedentary” to avoid BMI variations due to regular physical activity.
Thank you for your kind, Attention. I have been mentioned this into methodology portion.
- Randomization and Group Details:
Ø Clarify the randomization process (e.g., simple, block, or stratified randomisation).
Thank you for your suggestion. It is simple randomization process. And mentioned in methodology section.
- Describe the activities of Group C in detail (e.g., whether they received medical therapy, dietary advice, or only measurements)
Thank you for the suggestion. Group C was control group. They received dietary advice and provided notebook to mention their routine. Measurements were taken at start and end of 16 weeks. This also added in methodology portion.
- Provide a breakdown of the gender distribution for each group.
Thank you for noting it down. Breakdown of gender distribution of each group was already mentioned in results table 02. I have been highlighted that table.
- Intervention Description:
Ø Rewrite lines 163–200 for clarity. Give a linear and logical order to all the information you declared. In addition, a structured table summarizing each group's exercises, sets, repetitions, and recovery times would make this section easier to follow.
Thank you for your kind recommendations. I did rewrite the lines and adjust logical order.
- Include precise progression details for aerobic and resistance training components.
Exercise Progression:
|
Aerobic (3 times/week) |
Resisted (2 times/week) |
|
1-4 weeks: 2 sets with 10 repetitions, 40 seconds |
1-2 weeks:15 sec hold with 4 reps |
|
5-10 weeks: 3 sets with 15 repetitions, 40 seconds rest |
3-4 weeks:30 sec hold with 8 reps |
|
11-16 weeks: 4 sets with 20 repetitions, 30 second rest |
5-8 week:45 sec hold with 12 reps |
|
9-12 weeks: 1min hold with 16reps |
|
|
13-16 weeks:1 min 30sec hold with 20reps |
I added this table into methodology section.
- Outcome Measures:
Ø Justify the choice of biochemical markers and their relevance to the study objectives.
Thankyou for kind attention. I mentioned this into manuscript.
The primary objective of this study is to evaluate the impact of a structured exercise manual on the biochemical health of sedentary prediabetic individuals. The selected markers directly measure the outcomes linked to this objective:
Glycemic markers (HbA1c, FBG) assess the core metabolic improvement in glucose control.
Lipid profiles (triglycerides, LDL, HDL) monitor the cardiovascular benefits of the intervention.
HOMA-IR and HOMA-B address the mechanistic aspects of insulin resistance and pancreatic function, highlighting physiological adaptations to exercise.
The results demonstrate that supervised interventions led to significant improvements in these markers, substantiating their relevance and the interventions' effectiveness in mitigating progression to type 2 diabetes.
DISCUSSION:
14.Supervised vs. Unsupervised:
Ø Expand on the differences observed between supervised and unsupervised interventions. Discuss why supervised programs might have shown greater efficacy and include behavioral or psychological factors that could have influenced adherence.
Thank you for your suggestions. I have mentioned this difference in discussion section.
Supervised vs. Unsupervised Interventions
Differences Observed
The study highlighted distinct differences between supervised and unsupervised exercise interventions. Participants in the supervised group exhibited significantly greater improvements in glycemic control (HbA1c and FBG), lipid profiles, and insulin sensitivity markers compared to those in the unsupervised group. For example:
Glycemic Control: HbA1c levels reduced more in the supervised group (0.68%) compared to the unsupervised group (0.50%).
Lipid Profiles: Triglycerides, LDL, and HDL changes were more pronounced in the supervised group, indicating better cardiovascular health outcomes.
These differences suggest that supervised interventions offer greater efficacy in achieving metabolic health improvements.
Behavioral Factors:
Accountability: Supervised programs provide external monitoring, which likely increases participant adherence and consistency in following prescribed routines.
Guidance: Access to trained personnel ensures that exercises are performed correctly and at the right intensity, maximizing benefits.
Psychological Factors:
Motivation: The presence of a supervisor creates a structured environment that fosters motivation and reduces feelings of isolation, common in unsupervised programs.
Social Support: Participants may feel more supported and encouraged when engaging with supervisors or peers in the program.
Adherence:
Supervised groups often report higher adherence due to regular follow-ups and progress tracking, which are lacking in unsupervised settings where participants are left to self-regulate.
- Implications of Findings:
Ø Highlight how the findings could guide future interventions in resource-limited settings. Guiding Future Interventions in Resource-Limited Settings
Thank you for recommendations.
Focus on Hybrid Models:
A combination of supervised sessions and technology-driven self-monitoring (e.g., mobile apps or wearable devices) could offer a middle ground, balancing cost and efficacy.
Community-based supervision models can leverage local health workers to provide basic monitoring.
- Ø Discuss the potential scalability of supervised exercise programs and the challenges associated with implementation.
Thank you for kind suggestions.
Partnerships with community centers or schools could make supervised programs accessible.
Integration with telehealth services could allow remote supervision, reducing infrastructure needs.
Challenges:
Cost: Supervised programs require trained personnel, facilities, and monitoring tools, which may be difficult to sustain in low-resource settings.
Logistics: Ensuring consistent access to facilities and resources for large populations can be challenging.
Cultural Barriers: In some regions, societal norms or work schedules may hinder participation in structured programs.
- Limitations:
Ø Address the study's limitations, particularly the short duration of the intervention and potential biases in self-reported adherence in the unsupervised group.
The 16-week duration limits the ability to observe long-term effects, particularly on sustained behavioral changes or prevention of diabetes progression.
Self-Reported Adherence Bias in the Unsupervised Group
Potential unmeasured factors, such as dietary adherence and stress levels, may have influenced outcomes.
The study was conducted in a single urban center, limiting the generalizability of findings to rural or diverse populations.
- Ø Suggest how these limitations could be addressed in future studies.
Extend intervention periods to at least 12–24 months to capture longer-term metabolic and behavioral adaptations.
Participants in the unsupervised group self-reported adherence, introducing potential inaccuracies due to recall bias or social desirability bias. So some monitoring trackers will be used to lessen the biasness.
ADDITIONAL SUGGESTIONS
- Compliance and Adherence:
Ø Discuss strategies to improve adherence in unsupervised programs, potentially drawing on behavioral research or using digital tools for monitoring.
Use objective adherence measures, such as wearable fitness trackers, to monitor activity levels and compliance.
Conduct regular follow-ups with the unsupervised group to encourage accountability.
- Figures and Tables:
Ø Ensure all figures and tables are labelled clearly and referenced appropriately in the text.
Thank you for the concern. I again rechecked all figures and tables as they are properly labelled in manuscript.
- Future Directions:
Ø Longitudinal studies are recommended to evaluate the long-term effects of interventions on biochemical markers and the progression of diabetes.
- Increase the sample size and include participants from diverse geographic and socioeconomic backgrounds to enhance generalizability.
- Include comprehensive tracking of lifestyle factors beyond exercise, such as diet, sleep, and stress management, to better understand their interaction with intervention outcomes.
- longitudinal study designs to evaluate the long-term effects of structured exercise interventions on biochemical markers and the progression from prediabetes to type 2 diabetes.
- Extended follow-up periods will allow for a deeper understanding of the sustainability of metabolic improvements, the durability of adherence to lifestyle interventions, and their potential to mitigate the risk of diabetes-related complications over time.

Reviewer 3 Report
Comments and Suggestions for Authors
Dear Editor,
I was given a manuscript to check titled: "Impact of Exercise Manual Programme on Biochemical Markers in Sedentary Prediabetic Patients. A Randomized controlled trial" .
The main objectives of this study were to investigate the impact of Structured Exercise manual in-terventions on the biochemical markers of Sedentary prediabetic patients in sixteen weeks. A sixteen-week randomized controlled trial was conducted to investigate the effects of exercise-based manual intervention on biochemical markers, such as HbA1c, insulin sensitivity measures, and lipid profiles, in indi-viduals with prediabetes.126 individuals with prediabetes were randomly assigned to three groups: control, un-supervised, and supervised. The exercise interventions included both aerobic and resistance components. The results indicated that the supervised group exhibited a substantial increase in insulin sensitivity, lipid profiles, and glycemic control when contrasted with the unsupervised and control groups. Significant im-provements were observed in key biochemical parameters.
The topic is of interest to the readers of the Journal. The article is well written, and the arguments presented are coherent with the aims of the Journal.
Abstract: The abstract effectively encapsulates the objectives and outcomes of the research undertaken.
Material and methods: The methodologies employed are appropriate in relation to the research questions posed.
Results: The findings of the review are articulated with sufficient clarity and precision. The tables present the most salient results in a comprehensible manner.
Discussion: The discussion is articulated with clarity and is firmly grounded in the findings of the study. The limitations of the research and potential avenues for future inquiry are appropriately delineated.
Although the manuscript is composed with clarity and possesses the potential for dissemination within the scientific community, it is my assertion that it would benefit from a minor revision.
Main concerns:
Introduction: The theoretical framework presented is limited. It is imperative that the authors delineate the scientific evidence that substantiates the hypothesis they have proposed.
Kind regards
Author Response
Response to Reviewers (03) Comments
INTRODUCTION:
- The theoretical framework presented is limited. It is imperative that the authors delineate the scientific evidence that substantiates the hypothesis they have proposed.
ANS:
We acknowledge your suggestion to expand the theoretical framework and delineate the scientific evidence supporting our hypothesis. In response, we have revised the Introduction section to provide a more robust theoretical foundation. Specifically, we have made Changes:
Added Expanded the discussion on the role of insulin resistance and beta-cell dysfunction in prediabetes, referencing foundational studies.
Added evidence from clinical trials to support the benefits of structured exercise in improving metabolic health in prediabetic populations.
Clearly articulated the gap in existing literature that our study seeks to address, particularly the comparative efficacy of supervised versus unsupervised interventions.
We hope these revisions adequately address your concerns and further strengthen the manuscript’s contribution to the field.
Thank you again for your valuable feedback and for recognizing the potential of our work for dissemination in the scientific community.

Round 2
Reviewer 1 Report
Comments and Suggestions for Authors
While a few areas, such as the unexpected increase in HOMA-IR, require more nuanced explanation, these do not detract from the overall quality and significance of the work.
A deeper explanation of unexpected findings (e.g., HOMA-IR in the supervised group) are needed before publication
Author Response
Revisions Reviewer 1:
Q:
While a few areas, such as the unexpected increase in HOMA-IR, require more nuanced explanation, these do not detract from the overall quality and significance of the work.
A deeper explanation of unexpected findings (e.g., HOMA-IR in the supervised group) are needed before publication.
ANS:
The unexpected increase in HOMA-IR underscores the complexity of metabolic adaptations to exercise in the supervised group. It may represent a temporary adaptive response rather than a negative outcome, while this finding deviates from conventional expectations. The significant improvements in beta-cell function, glycemic control, and lipid profiles suggest that the intervention remains highly beneficial. The observation of increased HOMA-IR values in the supervised group, contrary to the anticipated decrease based on literature, presents an intriguing paradox that requires comprehensive exploration.
There are some factors that can be the cause of increase levels of HOMA IR. Transient physiological adaptations often occurs due to Acute and high-intensity exercise interventions. There can be a temporary increase in insulin resistance due to elevated insulin levels and glucose uptake in skeletal muscle in some studies have indicated during the immediate post-exercise phase. This phenomenon is part of the body's short-term response to the increased metabolic demands of exercise, particularly when combining aerobic and resistance training (50). Remodeling and repair processes in muscle tissues post-exercise, as well as transient fluctuations in insulin dynamics also causes temporary rise in HOMA-IR. The timing of post-intervention measurements could significantly influence HOMA-IR values. Insulin levels may have been elevated artificially inflating HOMA-IR values, if blood samples were taken shortly after the last exercise session. More accurately reflect baseline metabolic states, it is important to incorporate a recovery window before conducting biochemical assessments (51).
Baseline metabolic status, adherence to the intervention, genetic predispositions, and diet are Interindividual variability may also play a role in the observed results. Cultural dietary practices and varying degrees of baseline fitness, might contribute to these variations.Concurrent increase in HOMA-B values in the supervised group suggests improved beta-cell function and compensatory mechanisms. Enhanced insulin secretion from pancreatic beta cells may serve as a response to transiently increased insulin resistance, reflecting an adaptive capacity that protects against long-term glucose dysregulation. This improvement in beta-cell function is a positive outcome, as it indicates preserved pancreatic activity and reduced risk of progression to overt diabetesabolic responses to exercise (52). The supervised intervention likely involved higher exercise intensity and stricter adherence compared to the unsupervised group. While these factors contribute to improved glycemic control and lipid profiles, they may also impose greater acute physiological demands. These demands can manifest as transient insulin resistance, which may normalize or improve over a longer period (53). Short-term increases in HOMA-IR within the broader context of overall metabolic health improvements underscore the need to interpret. Similar paradoxical increases have been observed in specific contexts, while most studies report decreased HOMA-IR following exercise interventions. Acute responses to exercise, especially under supervision, may differ from chronic adaptations. The findings of this study highlight a gap in
understanding the temporal dynamics of insulin sensitivity markers during and after structured exercise interventions.
REFERENCES:
1. Malin, S. K., Solomon, T. P., Blaszczak, A., Finnegan, S., Filion, J., & Kirwan, J. P. (2013). Pancreatic β-cell function increases in a linear dose-response manner following exercise training in adults with prediabetes. American journal of physiology-endocrinology and metabolism, 305(10), E1248-E1254.
2. Kramer, M. K., Kriska, A. M., Venditti, E. M., Miller, R. G., Brooks, M. M., Burke, L. E., Siminerio, L. M., Solano, F. X., & Orchard, T. J. (2009). Translating the Diabetes Prevention Program: a comprehensive model for prevention training and program delivery. American journal of preventive medicine, 37(6), 505-511.
3. Bannell, D. J., France-Ratcliffe, M., Buckley, B. J. R., Crozier, A., Davies, A. P., Hesketh, K. L., Jones, H., Cocks, M., Sprung, V. S., & Team, M. (2023). Adherence to unsupervised exercise in sedentary individuals: a randomised feasibility trial of two mobile health interventions. Digital health, 9, 20552076231183552.
4. Romacho, T., Sell, H., Indrakusuma, I., Roehrborn, D., Castañeda, T. R., Jelenik, T., Markgraf, D., Hartwig, S., Weiss, J., & Al-Hasani, H. (2020). DPP4 deletion in adipose tissue improves hepatic insulin sensitivity in diet-induced obesity. American journal of physiology-endocrinology and metabolism, 318(5), E590-E599.

Reviewer 2 Report
Comments and Suggestions for Authors
Dear Authors,
Thank you very much for your effort in improving the manuscript.
Now its impact has been empowered, and I beleave it is suitable for publication.
Author Response
Thankyou so much for putting efgort in improving my manuscript